# Virtual Dementia-Friendly Communities (Verily Connect) Stepped-Wedge Cluster-Randomised Controlled Trial: Improving Dementia Caregiver Wellbeing in Rural Australia

**DOI:** 10.3390/geriatrics8050085

**Published:** 2023-08-25

**Authors:** Irene Blackberry, Tshepo Rasekaba, Debra Morgan, Kayla Royals, Jennene Greenhill, David Perkins, Megan O’Connell, Mohammad Hamiduzzaman, Margaret Winbolt, Ainsley Robinson, Hilary Davis, Clare Wilding

**Affiliations:** 1John Richards Centre for Rural Ageing Research, La Trobe Rural Health School, La Trobe University, Wodonga, VIC 3689, Australia; t.rasekaba@latrobe.edu.au (T.R.); k.royals@latrobe.edu.au (K.R.); c.wilding@latrobe.edu.au (C.W.); 2Care Economy Research Institute, La Trobe University, Wodonga, VIC 3689, Australia; 3Canadian Centre for Rural and Agricultural Health, University of Saskatchewan, Saskatoon, SK S7N 5E5, Canada; debra.morgan@usask.ca; 4Faculty of Health, Southern Cross University, Bilinga, QLD 4225, Australia; jennene.greenhill@scu.edu.au; 5School of Medicine and Public Health, University of Newcastle, Newcastle, NSW 2300, Australia; david.perkins@newcastle.edu.au; 6Mental Health Policy Unit, Health Services Research Institute, University of Canberra, Canberra, ACT 2617, Australia; 7Department of Psychology, University of Saskatchewan, Saskatoon, SK S7N 5A5, Canada; megan.oconnell@usask.ca; 8Faculty of Medicine and Health, The University of Sydney, Sydney, NSW 2006, Australia; mohammad.hamiduzzaman@sydney.edu.au; 9Australian Institute for Primary Care and Ageing, La Trobe University, Melbourne, VIC 3086, Australia; margaret.winbolt@latrobe.edu.au; 10Goulburn Valley Health, Shepparton, VIC 3630, Australia; ainsley.robinson@gvhealth.org.au; 11Social Innovation Research Institute, Swinburne University of Technology, Melbourne, VIC 3122, Australia; hdavis@swin.edu.au

**Keywords:** caregiver, community, dementia, mobile application, rural, virtual, age friendly

## Abstract

Caring for people living with dementia often leads to social isolation and decreased support for caregivers. This study investigated the effect of a Virtual Dementia-Friendly Rural Communities (Verily Connect) model on social support and demand for caregivers of people living with dementia. The co-designed intervention entailed an integrated website and mobile application, peer-support videoconference, and technology learning hubs. This mixed-methods, stepped-wedge, cluster-randomised controlled trial was conducted with 113 participants from 12 rural communities in Australia. Caregiver data were collected using MOS-SSS and ZBI between 2018 and 2020. The relationship between post-intervention social support with age, years of caring, years since diagnosis, and duration of intervention were explored through correlation analysis and thin plate regression. Google Analytics were analysed for levels of engagement, and cost analysis was performed for implementation. Results showed that caregivers’ perception of social support (MOS-SSS) increased over 32 weeks (*p* = 0.003) and there was a marginal trend of less care demand (ZBI) among caregivers. Better social support was observed with increasing caregiver age until 55 years. Younger caregivers (aged <55 years) experienced the greatest post-intervention improvement. The greatest engagement occurred early in the trial, declining sharply thereafter. The Verily Connect model improved caregivers’ social support and appeared to ease caregiver demand.

## 1. Introduction

Living with dementia presents difficulties for both individuals living with dementia and their family and friends. Alzheimer’s disease, which constitutes the majority of dementia cases at 43.5% prevalence [1], is a progressive condition with no cure and is ultimately life-limiting. Caregivers often struggle with the emotional and pragmatic challenges of ongoing losses of ability in people living with dementia. As the condition progresses, the care needs of people living with dementia also increase. Caregivers often face the demanding task of providing extensive care, which can lead to an increased risk of social isolation, as they have less time and energy for socialising [2,3]. The time needed to provide care and the emotional toll of caring can mean that caregivers of people living with dementia experience a reduced amount of time for family and friendships and often forego holidays and hobbies [4]. As a result, caregivers may find that their social support network gradually diminishes [5].

Living with dementia in rural communities presents additional demands and challenges for people living with dementia and their caregivers. As the rural population continues to age at an unprecedented rate [6], the incidence of dementia is also increasing due to the well-established association between ageing and dementia [7]. This places additional strain on already resource-limited healthcare and social services. Rural older adults may face challenges in accessing care services and often must travel long distances to reach them [8,9,10]. Furthermore, caregivers and providers in rural areas have expressed frustration with the fragmented health system’s limited ability to provide tailored services that meet the specific needs of dementia caregivers [9,11]. In small rural communities, where residents are often closely acquainted, dementia may attract social stigma [12], leading caregivers to avoid seeking help and maintaining privacy to avoid the associated stigma [9]. 

Despite the existing challenges in delivering support to individuals with dementia and caregivers in rural areas, the emergence of information and communication technologies (ICT), such as mobile applications (apps), videoconferencing, and social media, have opened fresh possibilities to address the accessibility gaps in services and social connections. These online technologies offer potential benefits by bridging geographical distances that currently hinder access to services and support [13]. Multiple studies have examined eHealth strategies and have demonstrated the effectiveness of computer-mediated interventions in assisting informal caregivers of individuals with dementia. These interventions have shown positive outcomes in alleviating caregiver depression, anxiety, demand, and stress and enhancing self-efficacy, knowledge, and skills [14,15,16]. Technology-based interventions also offer added advantages, such as caregiver anonymity and convenient access at any time [15].

Although advances in technology provide opportunity for connecting caregivers of people living with dementia to other caregivers, and for supporting service providers and other community organisations to become more dementia-friendly, there is a lack of evidence about how to deliver these outcomes within heterogenous Australian rural communities. To address this gap, 12 Australian rural communities of caregivers of people living with dementia, aged care and health providers, and volunteers worked with researchers from 5 universities to co-design a system of web-based information and support called Verily Connect (a loose acronym for Virtual Dementia-Friendly Rural Communities). The Verily Connect model involved providing online resources (the Verily Connect App and facilitated videoconference meetings) and location-specific resources (localised information, training for volunteers, and Verily Connect Hubs) to build both online and geographically specific dementia-friendly communities. The volunteers provided support to caregivers, enabling the caregivers to engage with Verily Connect technologies. Verily Connect volunteers’ perspectives are published elsewhere [17], and aged care and health provider experiences will be reported separately. This paper presents an evaluation of the Verily Connect model on caregivers’ perceived social support, demand, and use of the Verily Connect technology. Additionally, we examined the cost of implementing the Verily Connect model.

## 2. Materials and Methods

### 2.1. Study Design

The Verily Connect model implementation study was a stepped-wedge, open-cohort cluster-randomised controlled trial [18,19]. The study protocol detailing the procedures for this pragmatic evaluation of a co-designed intervention involving multiple stakeholders is published elsewhere [20]. With each of the 12 participating rural communities making up a cluster, the Verily Connect model was implemented progressively across 3 clusters at a time. Each implementation step lasted eight weeks (Figure 1). The implementation was completed over four steps spanning from 2018 to 2019, resulting in a total implementation period of 32 weeks. 

There are three main components of the co-designed Verily Connect model, including:-An integrated website and mobile app (Verily Connect app).-Volunteer support and a Technology Learning Centre (Verily Connect Hub) that was physically located in each rural community.-Caregiver peer support groups that met via web-based videoconference.

The Verily Connect website and mobile app contained 12 curated evidence-based guides for caregivers (Figure 2). The guides were brief and could be accessed and completed at any time and pace. Website sources and a directory of local dementia-relevant services and resources were displayed using Google Maps. Service information included links within the app that directly connected app users to the telephone, email, Facebook link, and website of the listed service.

### 2.2. Ethics Approval

This study adhered to ethical and legal requirements outlined in the National Statement on Ethical Conduct in Human Research (2007) and followed the principles of the Helsinki Declaration. Approval was granted by the Melbourne Health Human Research Ethics Committee (HREC/17/MH/404, reference 2017.376). 

### 2.3. Trial Registration

The trial was registered with the Australian New Zealand Clinical Trials Registry ACTRN12618001213235; https://tinyurl.com/4rjvrasf (accessed on 19 August 2023). and the International Registered Report Identifier (IRRID) is RR1-10.2196/33023. 

**Figure 2 geriatrics-08-00085-f002:**
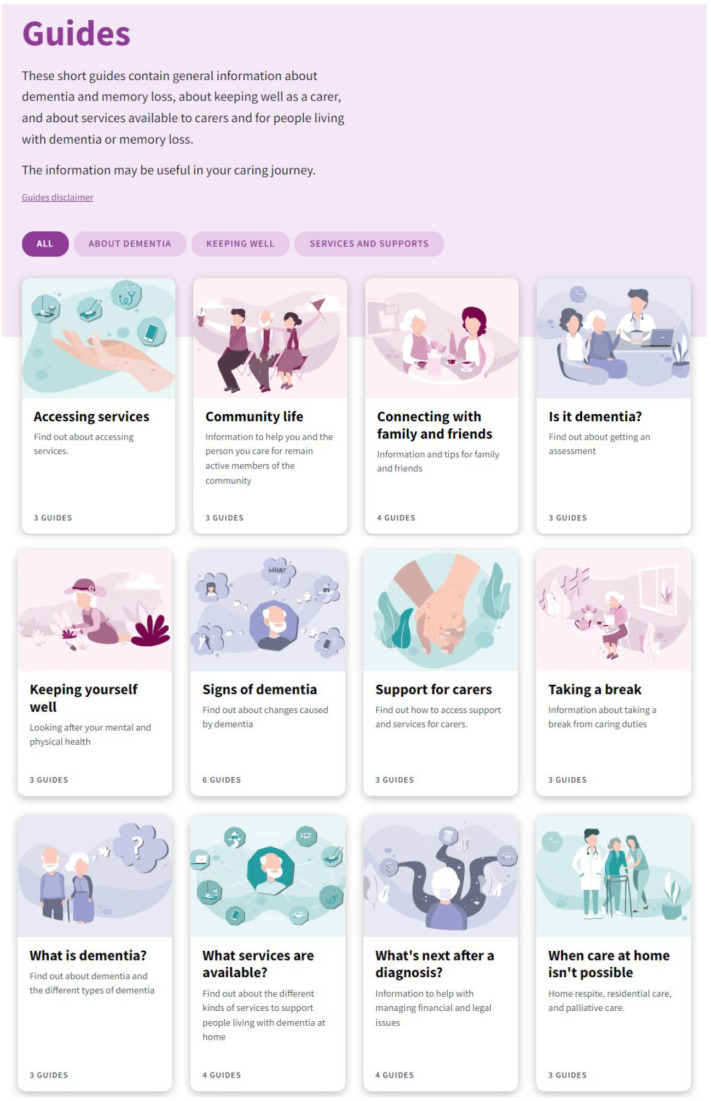
Example of guides.

### 2.4. Study Setting and Sample

A total of 113 participants were recruited, including 37 caregivers, 39 volunteers, and 37 health service staff. Participants were recruited from 12 rural communities in Australia, including 8 from the state of Victoria (Edenhope, Warracknabeal, Heathcote, Horsham, Kyneton/Macedon Ranges, Robinvale, Koo Wee Rup, and Mansfield), 2 from New South Wales (Molong and Nyngan), and 2 from South Australia (Riverland and Victor Harbor). These states share borders and are in the southeastern corner of Australia. Although cluster sampling was randomised, the recruitment of individual participants involved convenience sampling. Although the Verily Connect website and app were available on public platforms, the study project management team had control over who could sign up and access the app, thus restricting access to only enrolled participants. 

### 2.5. Recruitment

A local rural health service partner was identified in each participating community, and local health service staff were enlisted to help promote the study and identify potential participants for direct recruitment by the research team. Various recruitment strategies were employed, including open community forums, distribution of advertising materials, meetings with community organisations and groups, use of social media and media releases, paid advertising in newspapers and radio, an online project launch, and use of the “Contact Us” form on the Verily Connect app webpage [21]. 

### 2.6. Data Collection and Analysis

Caregivers completed surveys during six time-windows spaced six weeks apart, with each window lasting two weeks. The first survey was conducted immediately before the first implementation step, and the final survey took place during a follow-up period five months after the last implementation step was completed. Four survey rounds, each comprising three surveys, and two survey rounds, each comprising just one survey, were conducted. The three surveys were the Medical Outcomes Study Social Support Survey (MOS-SSS) [22], the Zarit Burden Interview (ZBI) [23], and bespoke surveys that collected demographic and background information. The MOS-SSS was used for the single survey rounds. The range of possible total scores for the MOS-SSS raw scores was 0 (lowest social support) to 100 (greatest social support), with scores <50 indicating a need for social support [22]. The ZBI is scored on a scale ranging from 0 (no demand) to 88 (severe demand) [23]. Mean imputation was used for missing survey data for cases lost to follow-up. Loss to follow-up is common, decreases study power and internal validity of a trial, and may lead to bias in reporting a trial outcome [24]. Creating mean data of those lost to follow-up enabled informative analysis to be performed to answer the research question and minimized bias [25,26].

The pre- and post-Verily Connect model implementation caregiver outcomes, overall social support (MOS-SSS) and caregiver demand (ZBI) scores, were compared using paired samples *t*-tests. Correlational analyses of social support and caregiver demand with caregiver age, years of caregiving, duration of dementia for people living with dementia (since diagnosis), and duration of exposure to the Verily Connect model were explored. Further, exploratory regression analyses of the relationship between post-intervention social support scores and age, years of caregiving, years since dementia diagnosis, and intervention duration were performed using the smoothing spline method. The smoothing spline method is useful for modelling dose–response relationships in regression analyses with noisy or highly variable data [27,28].

User engagement with the Verily Connect app and website was investigated using Google Analytics reports, in which aggregated data were presented as a summary of website/app traffic, including traffic volume by devices used. In the analysis, “users” represent the total count of logins based on Internet Protocol addresses rather than unique individuals associated with the login session.

A cost analysis was performed, focusing on the initiation and implementation of the Verily Connect model, considering the perspectives of project managers, officers, and health services involved in the 12 trial communities. Data were collected on insights into the expenses associated with delivering the Verily Connect model. The cost estimates, presented in Australian dollars based on 2019 costings, encompassed various resources required for implementing and running Verily Connect, including recurrent costs, personnel time, materials and equipment, utilities, and space. Descriptive analyses of the economic evaluation are reported elsewhere [29]. 

## 3. Results

Caregiver characteristics are reported in Table 1. Fifty-one percent of the participants were enrolled prior to step two of the implementation period. However, there were additional enrolments with each implementation step, comprising participants from communities that were moving from the control to the implementation phase. Participant flow is reported in Figure 3. Pre-implementation primary outcomes surveys were completed by 27 caregivers. One participant received the intervention and did not return the baseline survey, although the participant returned subsequent surveys. Five surveys were returned at the five-month follow-up. 

### 3.1. Caregiver Perceived Social Support and Demand

The caregiver social support (MOS-SSS) scores exhibited a positive trend towards improvement in social support by Survey Round Three (week 16) (Table 2). Subsequently, the MOS-SSS scores slightly decreased in the following two survey rounds (weeks 24 and 32), although they still remained higher than the baseline level. Furthermore, caregiver demand, as assessed by the ZBI, showed marginal improvement by Survey Round Five (week 32). 

The significant improvement in MOS-SSS scores from baseline (Survey Round One) to the end of the implementation period (Survey Round Five, week 32) [mean (SD) difference = 10.8 (17.8), two-sided *p* = 0.003, Cohen’s d = 0.61] demonstrated significant social support over 32 weeks. Although caregiver demand decreased, indicating an improvement in the caregiver’s situation, this reduction did not reach statistical significance (ZBI mean (SD) difference = −4.5 (13.4), two-sided *p* = 0.09, Cohen’s d = −0.33). Only five participants completed the surveys at the five-month post-implementation follow-up, yielding the following mean (SD) scores: MOS-SSS = 47.6 (45.6), ZBI = 57.8 (6.5).

### 3.2. Effect of Age, Years in Caring Role, and Duration of Intervention Exposure Time (Days) on Social Support and Caregiver Demand 

Post-intervention social support was not significantly (*p* > 0.05) correlated with caregiver age (rs = −0.017), duration of caregiving (rs = −0.237), dementia duration (rs = −0.189), and duration of intervention exposure (rs = −0.093). Similarly, post-intervention caregiver demand was not significantly associated with caregiver age (rs = −0.312), duration of caregiving (rs = −0.076), dementia duration (rs = −0.089), or duration of intervention exposure (rs = −0.256).

The respective relationships between social support and caregiver demand and other related factors based on the smoothing spline method are illustrated in Figure 4 and Figure 5, respectively. Thin plate splines (heatmaps) are shown for pre-intervention social support and caregiver demand, respectively. The light green to yellow shades indicates high post-intervention scores, and the darker blue means low post-intervention scores.

Caregivers around age 55 with a pre-intervention social support score of 40 had the largest increase (20 points) in social support after the Verily Connect implementation (Figure 4a). The smoothing spline in Figure 4b indicates a change point at around 55 years old, suggesting that caregivers under 55 years experienced improved social support associated with the Verily Connect model, whereas those over 55 years of age experienced a decline in social support. Additionally, caregivers who had been providing care for a person living with dementia for one to three years and had a pre-social support score of approximately 40 (Figure 4c) appeared to be more likely to receive greater social support (20-point improvement) from engaging with the Verily Connect model, especially during the first year of caregiving (Figure 4d). The first two years of the caregiving relationship showed the greatest improvement in social support (MOS-SSS=50 to ~70–80 ), and there was another “improvement bump” seen at six–seven years (Figure 4e). The Verily Connect model had the most benefit for caregivers in the first year of their role (Figure 4f). Additionally, longer exposure to the model was associated with a greater improvement in social support (Figure 4g,h).

Figure 5a–h show the relationship between caregiver-reported demand (ZBI scores) and other variables. Caregivers aged 50 years experienced the highest demand (Figure 5a,b), and the sense of demand decreased steadily with increasing age (Figure 5b). Those people in the initial three years of caregiving reported experiencing the greatest demand, starting with a baseline ZBI score of 75 and decreasing to approximately 55 (Figure 5c). Furthermore, a longer caregiving experience was associated with reduced demand following the implementation of the Verily Connect model (Figure 5d). It would appear the duration of diagnosis for the care recipient had no impact on the caregiver’s demand or response to the implementation of Verily Connect (Figure 5e,f). There was no clear relationship between caregiver demand and exposure to the Verily Connect model (Figure 5g,h), although exposure longer than 125 days may be associated with a slight lessening of demand.

### 3.3. Engagement with Verily Connect Website/App

The analysis of Google Analytics reports revealed that the highest number of users and visits to the Verily Connect online technology occurred during Step 2 (October to December 2018), with a decline observed afterward (Figure 6), corresponding to participant attrition. The majority of users accessed the Verily Connect technology using desktop computers, as indicated in Table 3. However, the mobile app version of Verily Connect accessed through tablet devices exhibited higher engagement metrics, including that the number of pages visited was twice the rate for tablet users compared with desktop users, and the average time spent on a page was ten times longer for tablet users than access from a desktop.

### 3.4. Cost Estimates and Analysis

Table 4 provides an overview of the average costs associated with implementing the Verily Connect model based on 2019 costs. The high personnel costs primarily stemmed from extensive research rather than direct model delivery. By excluding personnel costs related to research project officers’ wages, the recurring costs for operating the Verily Connect model would be significantly reduced (Figure 7). Moreover, the one-time start-up cost per community averaged AUD 21,000, regardless of the number of participants in the community.

## 4. Discussion

### 4.1. Primary Results 

The Verily Connect study was the first pragmatic trial of a hybrid dementia-friendly community model to connect rural dementia caregivers to localised in-person and online support. At baseline, the study highlighted that informal caregivers of people living with dementia in rural Australia experience low levels of social support and high levels of demand. After the Verily Connect model was implemented, caregivers’ perception of the social support they received was improved, as evidenced by the increase in MOS-SSS scores at week 16 post-baseline, and these score improvements were sustained until week 32. Use of the model also produced positive trends towards less demand in caregiving over time as a result of the Verily Connect model as evidenced by the ZBI scores. As dementia is a progressive condition, the caring demand increases as the functioning of people living with dementia deteriorates, and these increases in caring requirements can reduce caregivers’ access to their social support network [5]. However, our study demonstrated that a technology-based intervention delivered in a hybrid mode could significantly improve the low caregivers’ levels of social support. Although there can be challenges for employing technology-based interventions, the trend toward reductions in ZBI scores observed as the Verily Connect model was implemented, indicated that although caring duties may have increased over time, the use of Verily Connect technology did not place an additional demand on caregivers and rather may have alleviated some of the struggle experienced by caregivers. In addition to the time requirements in learning and using a technology-based intervention, it is also known that older adults and rural communities often experience poor internet connectivity and reduced access to updates [30,31]. Through co-designing the Verily Connect model we aimed to mitigate these anticipated challenges, and thus we developed a flexible hybrid approach to supporting caregivers and to training volunteers and staff. We aimed to reduce impediments to technology use through making devices and access to the internet available at the local community level including at health services and libraries. Another known challenge for technology-based interventions is that usage attrition rates in technology-based interventions tend to be higher among older people [32]. However, even with the support of local health services and volunteers, and using processes of whole-of-community consultation and co-design, the attrition rate in our study was low. The modest results achieved in the study reflect the real-world, ongoing challenges of using web-based technology with rural older people who are providing care for people living with dementia, which is an illness with devastating consequences. There were some critical factors, such as internet connectivity, transport, staff turnover, and the health status of caregivers and their care recipients over time, that have impacted our study implementation.

This study identified potentially important relationships between caregiver age, years of caring, and caregiver outcomes. Results of mapping scores showed that caregivers around 55 years old and in their first one to three years of caring experienced the highest increase in perceived social support after using the Verily Connect model. This suggests that younger caregivers with less experience in caring may benefit from technology-based interventions more than older and long-term caregivers. These findings align with previous studies that reported greater engagement with technology-based interventions among younger caregivers [33,34]. For instance, a systematic review of internet-based interventions for dementia caregivers found that younger caregivers were more likely to participate in such interventions [33]. Similarly, a study of an interactive web tool for facilitating shared decision making in dementia-care networks found that participants aged 70 years or younger were more able to use the tool than those over 70 years [34]. These findings underscore the importance of tailoring technology-based interventions to the needs and abilities of different caregiver groups. 

This study also highlighted some of the barriers to the uptake of online-delivered support programs for rural dementia caregivers. One of the main obstacles to recruitment was that some older people in the target caregiver population preferred not to use online technologies, despite the absence of locally available face-to-face caregiver support groups. Although local Verily Connect volunteers were available to provide face-to-face assistance with accessing the online support, caregivers still preferred face-to-face caregiver support to the online option provided by Verily Connect. Moreover, caregivers tended to be time-poor and emotionally overworked, which may have hindered their willingness to take on an additional activity, even if it might benefit them. Some caregivers experienced poor internet connectivity in their rural areas, hindering full engagement with the support available through the Verily Connect model. Systematic reviews of technology-based interventions for informal dementia caregivers have highlighted recruitment challenges and low engagement with telehealth interventions [33,35]. To address these issues, one review suggested simplifying the interventions, providing access to support through health professionals, and emphasising the benefits [33]. Although our study implemented the recommended strategies, recruitment remained a challenge. Further research is needed to identify effective strategies for recruiting and engaging older caregivers in technology-based interventions for dementia.

Implementation of the Verily Connect model was an innovation and resulted in the creation of an additional support service in each rural community; it augmented and added to existing support services. The costing results reflect the expense incurred in establishing this new service. The primary cost associated with the Verily Connect model was related to the project manager’s and officers’ time organising, promoting, and developing the program in each community, which included significant travel expenses. This is the case with many health programs, where personnel costs of salaries and wages are the most significant expense [36]. However, future rollouts of the Verily Connect model may not require similar levels of travel. For example, training could be conducted online, and meetings could be held virtually. In a post-COVID-19 world, understanding and acceptance of videoconferenced meetings as “usual practice” is higher than when the study was undertaken. Additionally, now that the Verily Connect model has been well defined after the trial period, local collaborators living in the communities could play a more prominent role in hosting community events, serving as Verily Connect champions who provide face-to-face liaisons with caregivers and volunteers. In this scenario, travel costs would likely decrease significantly. Although the time contribution of local health services and community staff would likely increase, this would be offset by a decrease in project staff time. It is unknown if increases in time contributed to the implementation of the Verily Connect model by local health services’ staff might be offset by reduced demand on existing health services, as caregivers may experience increased support by participating in Verily Connect activities. Overall, these modifications could potentially reduce the cost of implementing the Verily Connect model in the future. 

We are unable to assess how the cost of the Verily Connect model compares with other technology-based caregiver interventions because there is a lack of studies that have completed high-quality economic evaluations [37]. Many studies in this area report cost-effectiveness in terms of whether or not the intervention reduced the use of social and care services by the caregiver and person living with dementia (i.e., cost to society) rather than the cost of implementing the intervention [37,38]. Within the total costs, there are developmental/set-up costs that are negligible once the program is implemented. Of the studies that do report implementation costs [38], it is not possible to directly compare costs due to the heterogeneity of programs. 

Overall, the findings of our study emphasise the significance of offering support and resources to caregivers of people living with dementia in rural communities. Implementing technology-based interventions can be an effective approach to enhancing social support for caregivers and mitigating some of the obstacles associated with providing services in rural areas. Nonetheless, this study highlights the necessity of customising interventions to the unique requirements and characteristics of the caregivers. By offering tailored, well-supported interventions to caregivers in rural areas, the quality of life and wellbeing of people living with dementia and their caregivers may be enhanced, ultimately promoting a more inclusive and equitable healthcare system for all. 

### 4.2. Study Strengths and Limitations 

This study provides evidence about rural communities that were not exposed to the use of technology, as it was conducted prior to and until the beginning of COVID. This study thereby offers a unique insight into engaging with older people in rural communities, including co-designing a dementia-friendly program that has a face-to-face component and a virtual online component. The co-design is important to make the Verily Connect program sustainable. This Verily Connect program is currently being expanded to be launched nationally in late 2023. This study produced a toolkit to assist other communities in implementing the Verily Connect model in their own locality. The strength of this trial is that we used a highly regarded stepped-wedge cluster-randomised controlled trial design and economic evaluation. 

The study outcomes may have been influenced by various factors, including significant challenges in recruiting study participants in rural areas. In addition, this study experienced further challenges associated with the recruitment and sustained participation of caregivers of people living with dementia [39]. Significant resources were required to promote the program, recruit caregivers, and sustain their participation (in particular, travel costs). Unfortunately, the sample size for the caregiver cohort fell short of the desired number, which could limit the generalisability of the findings. Another limitation of this study was the poor response rate at the five-month follow-up after Verily Connect implementation. This reduced data prevented exploration of whether the trends observed during the 32-week trial period were sustained beyond the trial implementation period. As a result, the longer-term sustainability of the Verily Connect model’s effects remains uncertain. 

### 4.3. Future Studies

Despite the low number of caregiver participants, the current study demonstrated the potential of a hybrid of online technologies in supporting rural caregivers of people living with dementia, in addition to local community-based support. As online-delivered support services become more widely used and digital literacy continues to improve across the care sector, future cohorts of caregivers may be more accepting of online-delivered support services. The COVID-19 pandemic has resulted in a significant increase in the use of online-delivered services and further innovation and uptake of online-delivered services is likely to occur. As a result, in the future, there may be more people willing to accept programs that deliver services online and participate in studies that evaluate these programs. Additionally, it is essential to acknowledge that technology is continually evolving, and the Verily Connect model should also evolve in response to technological and social changes. A program of continuous improvement, evaluation, and development is therefore recommended. Post-COVID, as more people are exposed to technology, there will be a higher uptake of technological interventions. This is something that will be monitored in future iterations of the Verily Connect program, as well as the evaluation of its long-term effectiveness. 

## 5. Conclusions

This study illustrated that there is potential for using a technology-based intervention to increase perceived social support on rural caregivers of people living with dementia but without placing undue additional care demands. Despite limitations in sample size and participation in follow-up, this study highlighted the importance of providing tailored support and resources for caregivers in rural communities and it delivered more understanding about how caregivers’ demographic characteristics, preferences, and available resources influence the uptake of technology-based intervention. This study exemplified the potential of technology to overcome complex challenges associated with providing locally based support services in these rural areas and in connecting caregivers across Australia despite significant geographical barriers. Technology can offer flexibility for caregivers to access the support they require when they need it. Through allowing anonymity and privacy for caregivers in rural communities to seek help without having to disclose their personal identities online, web-based technology can also reduce some of the stigma experienced by rural caregivers. As digital literacy and accessibility of low-cost, scalable online-delivered service delivery continue to increase, the future holds the promise that technologies can be employed to cost-effectively deliver services for rural caregivers of older people living with dementia. Further research is needed to strengthen the evidence base in the use of technology for rural caregivers.

## Figures and Tables

**Figure 1 geriatrics-08-00085-f001:**
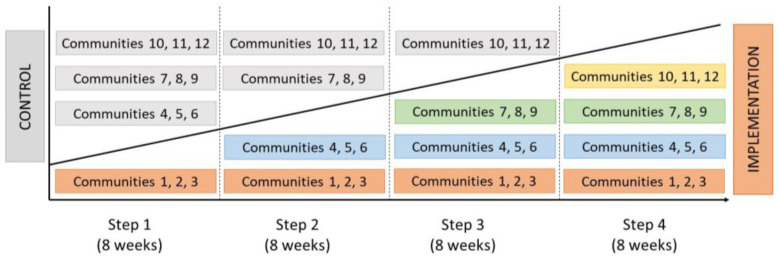
Stepped-Wedge Implementation of the Verily Connect Model across the 12 Clusters.

**Figure 3 geriatrics-08-00085-f003:**
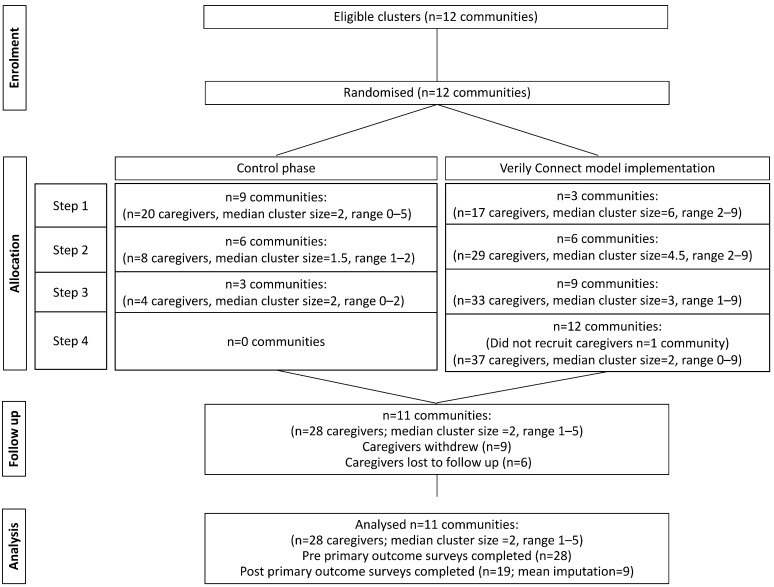
CONSORT Diagram Showing the Flow of Participants through the Trial.

**Figure 4 geriatrics-08-00085-f004:**
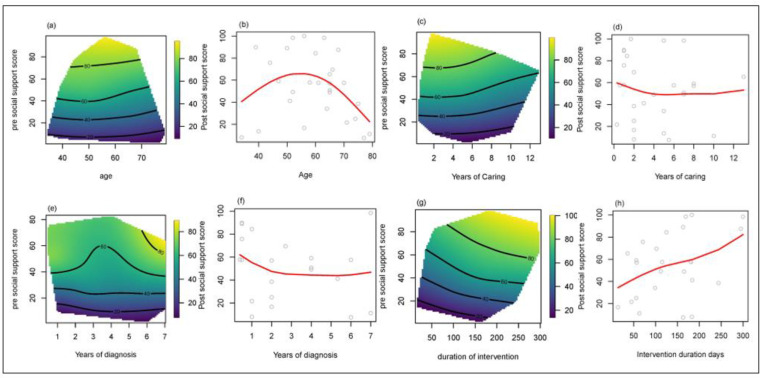
Smoothing Spline Visualisation of Relationship between pre- and post- Caregiver Demand and Other Factors. (**a**,**b**) Effect of age on caregiver social support. (**c**,**d**) Effect of years in a caregiving role on caregiver social support. (**e**,**f**) Effect of years since diagnosis on caregiver social support. (**g**,**h**) Effect of Verily Connect Model intervention duration on caregiver social support.

**Figure 5 geriatrics-08-00085-f005:**
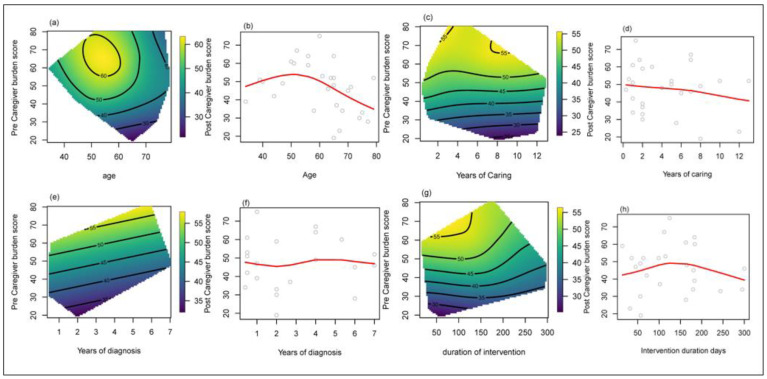
Smoothing Spline Visualisation of Relationship between pre- and post- Caregiver Demand and Other Factors. (**a**,**b**) Effect age on caregiver experienced burden. (**c**,**d**) Effect of years in a caregiving role on caregiver experienced burden. (**e**,**f**) Effect of years since diagnosis on caregiver experienced burden. (**g**,**h)** Effect of Verily Connect Model intervention duration on caregiver experienced burden.

**Figure 6 geriatrics-08-00085-f006:**
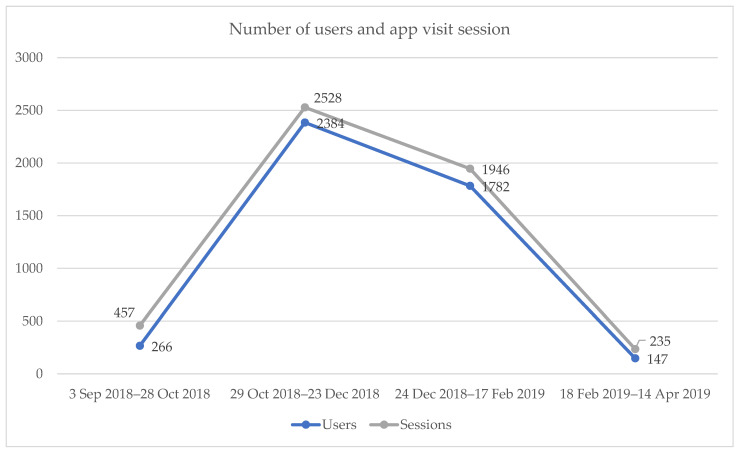
Volume of Website/app Traffic.

**Figure 7 geriatrics-08-00085-f007:**
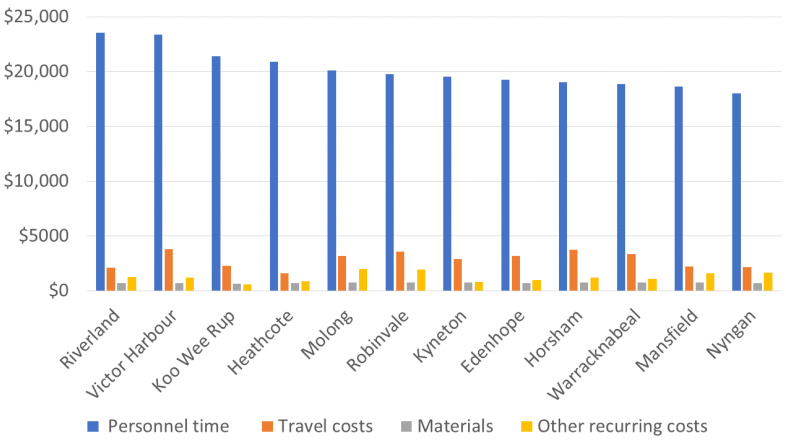
Breakdown of the Recurrent Costs in AUD ($).

**Table 1 geriatrics-08-00085-t001:** Caregiver Demographics.

Variables	Values
Age Mean (SD), years	60 (12)
Female sex, n (%)	32 (86)
Has a home care package, n (%)	5 (14)
Highest level of education, n (%)	
Secondary school (Year 7–Year 11)	13 (34)
Secondary school/TAFE/College	15 (41)
Undergraduate tertiary education	4 (10)
Postgraduate tertiary education	5 (14)
Years in caring role, n (%)	
<2	13 (35)
2–6	13 (35)
>6	11 (30)
Care recipient’s relationship to caregivers, n (%)	
Parent	18 (48)
Spouse	14 (38)
Sibling	4 (10)
Friend	1 (3)
Care recipient’s diagnosis, n (%)	
Dementia	27 (72)
Cognitive impairment	4 (10)
No formal diagnosis	7 (18)
Care recipient’s years since diagnosis, n (%)	
<2	13 (34)
2–4	5 (14)
>4	9 (24)
Caregiver’s location of residence, n (%)	
Lives with care recipient	17 (45)
Same postcode as care recipient	17 (45)
Lives 50–100 km from care recipient	4 (10)

**Table 2 geriatrics-08-00085-t002:** Mean (SD) Total MOS-SSS and ZBI Scores, n=28.

	Survey Round 1	Survey Round 2	Survey Round 3	Survey Round 4	Survey Round 5
MOS-SSS	43.2 (25.9)	48.4 (26.4)	54.0 (29.3)	56.1 (29.5)	54.1 (29.4)
ZBI	51.4 (14.7)		50.0 (15.3)		46.9 (13.8)

**Table 3 geriatrics-08-00085-t003:** Website/app Engagement by Access Device and Duration.

	No. of Logon Sessions (%)	No. of Visited Pages per Session	Average Session Duration, Seconds
Desktop	4664 (90.0)	1.63	52
Mobile	357 (6.9)	1.58	207
Tablet	163 (3.1)	3.01	485
Total	5184 (100)	1.67	67

**Table 4 geriatrics-08-00085-t004:** Average Costs for Implementing the Verily Connect Model.

Cost Category	Average Cost	Share of Total Cost
Personnel time	$20,203	77%
Materials	$703	3%
Travel costs	$2846	11%
Other recurrent costs	$1282	5%
Recurrent costs	$25,035	92%
Equipment	$797	3%
Furniture	$14	0%
Vehicles	$193	1%
Buildings	$1022	4%
Capital costs	$2026	8%
Total costs	$27,061	100%
Start-up costs	$21,036	

## Data Availability

The data presented in this study are available on request from the corresponding author. The data are not publicly available due to the privacy of participants.

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
