# Peer review of "Virtual Dementia-Friendly Communities (Verily Connect) Stepped-Wedge Cluster-Randomised Controlled Trial: Improving Dementia Caregiver Wellbeing in Rural Australia"

_geriatrics, 2023, doi:10.3390/geriatrics8050085_

Round 1

Reviewer 1 Report

It is  good article  involved , but to be acceptable in this journal.  There are minor corrections:

1- the abstract needs more clarification about current results 

2- the introduction needs more explanation about restricted subject.

3- there are no clear discussion for recent results in this study 

4- there is no comparison between current results and results in previous studies to improve current results 

5- also conclusion  is not enough for this study 

6- I accepted it after minor corrections 

Reviewer 2 Report

 The paper presents research based on a stepped-wedge cluster randomized controlled trial aiming to improve dementia caregiver wellbeing. This is an interesting work presented in a very detailed manner and each step is explained adequately. The methodological part is well performed and the paper overall is well written.

I considered it as a complete work presented in a proper way, including a comprehensible introduction and a clear description of the work done. It includes a lucid discussion and the limitations of the study.

Overall the findings have merit and the paper meets the standards for an international journal

I endorse publication in this form.

 it is fine

Author Response

Thank you for your positive and kind review of our paper. We sincerely appreciate your time and encouragement.

Reviewer 3 Report

Introduction included sufficient background to appreciate the demands on rural caregivers. The purpose of the study was clearly stated; however, the need for this particular study was not. The study design was clearly stated and was appropriate to address the stated purpose. It seems that the participants needed to be described in the methods section as they are not the results of the study. They were the people whose survey responses and usage patterns were tracked by the software. The results are presented clearly in the narrative. The tables provide good support for interpreting the narrative. In general, the figures add little to the clarity of the data presentation. In general, the discussion section is written well. The authors seemed to overemphasize the results that support their assertion, while providing minimal comment on the results that were not significantly different across the measurement times of the study. The authors mention attrition in the discussion, but needed to develop ideas on this concern more thoroughly,

This quantitative study provided data that supported the authors assertions for one of two measures, but not for the other one. It seems that a qualitative study with online interviews of the caregivers may have been a better method for investigating the research issue.

Introduction

Line 48 – the assertion made by the authors overstates the prevalence of Alzheimer’s disease among dementia types. The citation for the assertion was not a prevalence study. A better choice would be Goodman et al. (2017).

Line 75 – change the last word ‘has’ to ‘have’.

Lines 86-88 – The authors assert that research gaps exist, but do not indicate what those gaps are. They need to define the problem that the set out to address in the study.

Methods

Line 156 – The authors needed to explain the benefits vs. the challenges that came from including participants lost to follow-up and creating mean data for those responses that did not occur.

Line 164 – The authors needed to clearly state what were the ‘other related factors’.

Results

Line 187 – The authors needed to respond to the error message.

Line 188 and line 212 – The authors should remove the ‘n=’ from the narrative.

Line 190 – 18.5% response rate at five months. This level is lower than expected.

The data in Table 1 may be more clearly presented as count data as the base number of participants changed over the steps. These changes make it difficult for readers to interpret the data.

Figures 3 and 4 need clearer legends and axis definitions. The figures are explained clearly in the narrative.

Figure 5 – The lines imply that data were collected between the dots and that the changes were linear over time. Nothing in the Methods or Results section indicate that separate data collections occurred on a daily basis and that those data were aggregated. Therefore, the authors should remove the lines on the figure. Also, the data in this figure may be more clearly displayed as a table with four rows and three columns.

Line 276 – fix misspelled word.

Line 280 – The assertion in this line is not supported by the data. It may be that the people with tablets were more likely to spend extended time reading the information.

Figure 6 – Since the patterns were similar at all locations, this figure should be deleted. The similar pattern across locations can be described in the narrative.

Discussion

Lines 299-301 – The sentence needs to be rewritten to state the intent of the authors more clearly.

Line 311 – The authors indicate that the small sample size was a reason to feel heartened about the effect of their intervention. However, they cannot know how this small number of participants fit into the distribution of the target population.

Lines 312-315 – The authors present important concerns relevant to rural caregivers for people with dementia. However, they needed to discuss their ideas. They needed to develop these two sentences into a more thorough discussion of the concerns raised. The first concern is addressed again with more depth in the paragraph beginning on line 330.

Goodman, R., Lochner, K., Thambisetty, M., Wingo, T., Posner, S., & Ling, S. (2017). Prevalence of dementia subtypes in United States Medicare fee-for-service beneficiaries, 2011–2013. Alzheimer’s & Dementia, 13(1), 28-37. https://doi.org/10.1016/j.jalz.2016.04.002

Reviewer 4 Report

This paper describes the results of the Verity Connect trial to support carers in rural Australia. Since I live in a rural area in the UK with many of the issues discussed in the paper, I was interested in reading this paper.  Unfortunately it seems to me to be based on a report to funders rather than a scientific paper.  It did not tell me what support was provided as part of the project to the participating carers.  This may have been obvious to those involved in the project, but for me coming to the paper cold, as most readers will it was confusing.

I am not sure what the 'randomised trial' was intended to show.  It did seem to indicate an improvement in support, but I was unsure of either the baseline, were each cohort starting from a common baseline for example, and did the different timelines cause any issues?  It would help if the previous support and that provided in the trial were fully described and compared.

The financial information provided is only useful if compared with existing costs, which do not appear to be given.  Any healthcare provider wishing to test a similar system would want to know and understand the reasons for this difference.

Round 2

Reviewer 4 Report

The modifications have made this paper a lot clearer.  I can now see the objectives and how they were achieved.  It is now suitable for publication.

There is a dead link at line 215 and the formatting has gone awry after Figures 4 & 5.

Author Response

Dear Peer reviewer,

Many thanks for alerting us to the dead link. We have fixed this and attach a clean version where the formatting after Figure 4 and 5 is fixed. 

Best regards,

Irene
